# The Role of Mitochondria in Inflammation: From Cancer to Neurodegenerative Disorders

**DOI:** 10.3390/jcm9030740

**Published:** 2020-03-09

**Authors:** Sonia Missiroli, Ilaria Genovese, Mariasole Perrone, Bianca Vezzani, Veronica A. M. Vitto, Carlotta Giorgi

**Affiliations:** Department of Medical Sciences, Laboratory for Technologies of Advanced Therapies (LTTA), University of Ferrara, 44121 Ferrara, Italy; msssno@unife.it (S.M.); ilaria.genovese@unife.it (I.G.); prrmsl@unife.it (M.P.); vzzbnc@unife.it (B.V.); vttvnc@unife.it (V.A.M.V.)

**Keywords:** mitochondria, inflammation, neurodegenerative diseases, cancer

## Abstract

The main features that are commonly attributed to mitochondria consist of the regulation of cell proliferation, ATP generation, cell death and metabolism. However, recent scientific advances reveal that the intrinsic dynamicity of the mitochondrial compartment also plays a central role in proinflammatory signaling, identifying these organelles as a central platform for the control of innate immunity and the inflammatory response. Thus, mitochondrial dysfunctions have been related to severe chronic inflammatory disorders. Strategies aimed at reestablishing normal mitochondrial physiology could represent both preventive and therapeutic interventions for various pathologies related to exacerbated inflammation. Here, we explore the current understanding of the intricate interplay between mitochondria and the innate immune response in specific inflammatory diseases, such as neurological disorders and cancer.

## 1. Introduction

In addition to their role as cellular powerhouses by coupling metabolite oxidation through the tricarboxylic acid (TCA) cycle to the production of high amounts of adenosine triphosphate (ATP) by the electron transport chain (ETC), mitochondria are multifaceted organelles that execute a wide array of functions, including regulation of calcium (Ca^2+^) homeostasis, orchestration of apoptosis and differentiation [1].

Mitochondria contain their own DNA genome (mtDNA) that is expressed and replicated by nucleus-encoded factors imported into the organelle. mtDNA encodes thirteen proteins necessary for oxidative phosphorylation as well as the ribosomal and transfer RNAs needed for their translation. Mitochondria are functionally versatile organelles, which continuously elongate (by fusion), undergo fission to form new mitochondria or undergo controlled turnover (by mitophagy). These processes represent a fundamental framework of mitochondrial dynamics determining mitochondria morphology and volume and allow their immediate adaptation to energetic needs. Thus, mitochondria change their shape and number in response to physiological or metabolic conditions and guard against deleterious stresses that preserve cellular homeostasis.

Mitochondria are also the primary source of cellular reactive oxygen species (ROS) and are therefore highly involved in oxidative stress [2]. Under physiological conditions, ROS act as mitogen signals that provide many cellular functions; however, ROS overproduction leads to uncontrolled reactions with proteins, mtDNA and lipids, resulting in cell dysfunction and/or death.

In addition to their dynamic behavior in programmed cell death and metabolism, mitochondria are now considered central hubs in regulating innate immunity and inflammatory responses [3].

For decades, it has been observed that during the activation phase of an immune response, immune cells shift from a state of relative quiescence to a highly active metabolic state, which usually consists of a transition to a robust anabolic condition aimed at sustaining cell proliferation [4].

As a consequence, mitochondria have emerged as being necessary for both the establishment and maintenance of innate and adaptive immune cell responses [5,6]. Moreover, mitochondrial metabolism helps to control the function of immune cells beyond its role in generating ATP or metabolites that support macromolecule synthesis (for more details see [3])

Inflammation is a complex organismal response to infection and/or tissue damage in which various secreted mediators coordinate defense and repair and avoid further cell or tissue injury. Inflammation also stimulates tissue repair and regeneration to restore homeostasis and organismal health. Nevertheless, when the inflammatory process persists, it acquires new characteristics and drives various diseases in which inflammation and tissue damage/stress self-sustain each other [7].

An abundance of evidence points to a role for ROS generated by mitochondria (mROS) in regulating inflammatory signaling; thus, it is not surprising that mitochondria have been implicated in multiple aspects of the inflammatory response [8].

In general, inflammation induced by oxidative stress acts as a feedback system sustaining a stressful condition that could result in severe tissue damage and trigger chronic inflammation. This process is mainly orchestrated by the activation of the NOD-, LRR- and pyrin domain-containing 3 (NLRP3) inflammasome, currently the most fully characterized inflammasome [9] (see the text for further details). Thus, mitochondria can be considered the principal drivers of NLRP3-mediated inflammation as they can modulate innate immunity via redox-sensitive inflammatory pathways or directly activate the inflammasome complex. In line with this notion, (i) mitochondria represent a checkpoint of the intracellular cascades of numerous downstream pattern recognition receptors (PRRs); (ii) mitochondria have been recently identified as the major site for the generation of damage-associated molecular patterns (DAMPs), which are molecules recognized by the innate immune system that can polarize the fate of the inflammatory response by modulating the energetic level of immune cells in a NLRP3-dependent manner; and (iii) mtDNA has been implicated in NLRP3 inflammasome activation, inducing the release of proinflammatory cytokines so strongly that it can be considered a trigger of neurodegeneration [10].

In this review, we aim to discuss the role of the mitochondria in coordinating proinflammatory signaling, starting from their key role in modulating the innate immunity response and further focusing on mitochondrial dysfunction in two pathologies known to be supported and promoted by inflammation: neurodegenerative diseases and cancer. Nevertheless, we will briefly describe actual therapies aimed at targeting the mitochondria-driven inflammatory response.

## 2. Mitochondria: Key Players in Innate Immunity

In recent years, studies on mitochondrial control of immunity have expanded drastically, and they have increasingly identified mitochondria as key hubs in the innate immune system, acting as signaling platforms as well as mediators in effector responses. The first line of defense against dangerous stimuli is represented by the innate immune response [10]. Recognition of pathogens is predominantly arbitrated by a set of germline-encoded molecules on innate immune cells that are denoted to as PRRs. PRRs are able to perceive and distinguish conserved microbial structures, pathogen-associated molecular patterns (PAMPs) like lipoproteins, carbohydrate, microbial nucleic acids, or endogenous DAMPs including ATP, mtDNA, and cardiolipin, which are released by the cells of the host in response to injury or necrotic cell death [11]. 

These molecules often are similar to PAMPs both in terms of their structures and specific locations and can be exposed to PRRs during pathological conditions or failure of homeostasis. 

Among the PRRs, the membrane-bound Toll-like receptors (TLRs), nucleotide-binding oligomerization domain-like receptors (NLRs), C-type lectin receptors (CLRs), AIM2-like receptor (ALR) and retinoic acid-inducible gene I (RIG-I)-like receptor (RLR) activate the immune system and trigger a response against a pathogen, resulting in the activation of different intracellular signaling cascades, such as the activation of nuclear factor NF-kB (nuclear factor kappa-light-chain-enhancer of activated B cells) or the cellular kinase c-Jun amino-terminal kinase (JNK) [12], and the release of proinflammatory cytokines, chemokines and adhesion molecules, thereby accelerating the inflammatory response [13].

This section focuses on the key role of mitochondria in modulating innate immune responses after viral infection, NLRP3 inflammasome activation or bacterial exposure (Figure 1). 

### 2.1. Mitochondrial Dynamics During Viral Infection

Throughout viral infection, the immune response can be mediated by two receptors of the RLR pathway, melanoma differentiation–associated gene 5 (MDA-5) and the RIG-I, which detect cytoplasmic, virus-derived dsRNA [14]. These two pathways merge at the point of transcriptional activation, leading to the production of type I interferons (IFN-α and IFN-β). Notably, both RIG-I and MDA-5 contain two caspase recruitment domains (CARDs) that permit them to interact with the CARD domain of the mitochondrial antiviral signaling (MAVS) protein.

The identification of the MAVS protein initiated research on the role and function of mitochondria in the activation of innate immune pathways [15]. MAVS is embedded on the outer mitochondrial membrane (OMM) via its C-terminal transmembrane domain. MAVS is a pivotal signaling adaptor that activates NF-kB and IRF-3 signaling, inducing antiviral and inflammatory pathways for the production of proinflammatory cytokines and IFN-I during the development of innate immune responses to RNA viruses [16]. Intriguingly, it has been demonstrated that MAVS interacts with mitofusin-2 (MFN2) via a central 4,3 hydrophobic heptad region (HR1) [17]. In particular, a large amount of MFN2 sequesters MAVS in a nonproductive state, reducing both endogenous transcription factor interferon regulatory factor 3 (IRF-3) dimerization and NF-kB expression. On the other hand, the loss of endogenous MFN2 increases the production of IFN-β following a viral infection, which decreases viral replication. It is speculated that MFN2 inhibits dimerization at the CARD domain of MAVS [17]. Interestingly, similar results have not been obtained by modulating MFN1 expression levels, suggesting that MFN2 has a single role in regulating MAVS signaling, independent of its function in mitochondrial fusion. Noteworthy, cells deficient in both mitofusins lack the ability to undergo mitochondrial fusion and display a reduced MMP correlated to a defective cellular antiviral immune responses [18]. The dissipation of MMP has no effect on the activation of IRF-3 downstream of MAVS, suggesting that MMP and MAVS are involved in the same stage of RLR signaling pathway [18].

Moreover, MAVS interacts with stimulator of interferon genes (STING), a protein localized at the endoplasmic reticulum (ER) and involved in the antiviral cell response [19]. Upon infection with DNA viruses, STING is activated downstream of cGAMP synthase (cGAS) to induce IFN-I. STING interacts with RIG-1, stabilizing it, and activates both the IRF3 and NF-kB transcription pathways and the subsequent release of cytokines and proteins, such as the type I IFN, which exert its antipathogenic activities [20,21]. A growing body of literature has demonstrated that mitochondrial dynamics modulate antiviral RLR signaling, adding a new layer of complexity to mitochondrial antiviral immune responses [22,23]. 

Castanier and colleagues demonstrated that RLR activation promotes mitochondrial network elongation [24]. In an elegant way, they showed that MAVS binds MFN1 (functioning as a negative regulator of MAVS) and STING at the ER-mitochondrial interface, regulates mitochondrial morphology and facilitates the mitochondria–ER association required for signal transduction. Moreover, cytomegalovirus (CMV) infection impedes signaling downstream from MAVS and reduces the MAVS-STING association. [24]. Finally, they stated that mitochondrial fusion is required for efficient RLR signaling, as inhibition of fusion by knockdown of either MFN1 or optic atrophy 1 (OPA1) decreased virus-induced NF-kB and IRF3 activation [24]. On the other hand, cells depleted of dynamin-related protein 1 (DRP1) and FIS1, proteins involved in mitochondrial fission, displayed elongated mitochondrial networks and increased RLR signaling.

A recent study found that evolutionarily conserved signaling intermediate in Toll pathways (ECSIT) localizes at the mitochondrial surface with MAVS protein and mediates bridging of the MAVS protein to RIG-I or MDA5. In turn, ECSIT induces the activation of the antiviral response via upregulation of IRF3 and increasing the expression of IFN-β during viral infection [25].

Another example of how viral infection could modulate mitochondrial dynamics is represented by dengue virus (DENV) infection. DENV protease NS2B3 cleaves MFN1 and MFN2, which participate in host defense in different ways: MFN1 fosters MAVS-mediated IFN production and caspase activation, while MFN2 attenuates DENV-induced cell death acting on mitochondrial membrane potential (MMP) [26].

The findings regarding the role of proteins implicated in fission and fusion are controversial, but it is quite apparent that MFN1 and MFN2 interact with MAVS during RLR signaling, and further studies should clarify the exact roles of MFNs in this process. Taken together, these studies reveal as the mitochondrial fusion process is required for a proper ER-mitochondria connection and RLR-MAVS signalosome formation. 

Furthermore, upon viral infection, the mitochondrial calcium uniporter (MCU) complex amplifies the RLR signaling activation interacting with MAVS complexes at mitochondria and positively regulates the release of the proinflammatory cytokine IFN-β [27]. IFN-I production by the RLR pathway controls numerous IFN-stimulated genes (ISGs) by binding to interferon-α/β receptor 1 (IFNAR1) and activating downstream signaling via ER stress [27].

Emerging evidence has demonstrated that influenza A virus infection promotes mROS production, which drives innate immune inflammation and worsens viral pathogenesis. Pharmacological inhibition of mROS with the specific scavenger mitoTEMPO reduces airway and lung inflammation in an in vivo model and alleviates influenza virus pathology (31190565).

### 2.2. Mitochondrial Dynamics in NLRP3 Inflammasome Activation 

In the last few decades, inflammasomes have been increasingly recognized as playing an important role in innate immune and inflammatory responses. Among the several inflammasomes, the NLRP3 inflammasome has been the most studied and well characterized. The NLRP3 inflammasome is a multiprotein complex that consists of the scaffold protein NLRP3, the adaptor protein ASC (or PYCARD), and the enzyme caspase-1. NLRP3 is a molecular platform activated upon signs of cellular ‘danger’ to trigger innate immune defenses through the release of proinflammatory cytokines such as interleukin (IL)-1β and IL-18 [28]. 

A variety of endogenous and exogenous stimuli, such as extracellular ATP, microbial infection, bacterial pore-forming toxins and monosodium urate, asbestos and the adjuvant alum, are able to activate the NLRP3 inflammasome [29], but the mechanisms of activation are currently unclear. 

Furthermore, it has been proposed that ROS are responsible for NLRP3 inflammasome activation. The cellular sources of the ROS responsible for NLRP3 inflammasome activation need careful clarification, although numerous studies have convincingly excluded the NADPH oxidase isoforms NOX1, NOX2 and NOX4 [30,31]. Moreover, electron transport through mitochondrial oxidative phosphorylation (OXPHOS) is an important source of cellular ROS that, in turn, can damage cellular proteins, lipids, and nucleic acids via oxidation but can also be a critical second messenger in various redox-sensitive signaling pathways [32,33]. 

In this context, a great relevance has been attributed to the plethora of activities that takes place at mitochondria-associated ER membranes (MAMs), formed by the close apposition between the ER and mitochondria membranes (principally OMM) [34,35]. This defined region of the cell displays specific properties and a distinct set of proteins [36,37] that allows its purification by biochemical procedures [38]. It is well established that MAMs control several signaling pathways, from the regulation of lipid transfer to Ca^2+^ signaling [39], as well as coordination of ROS homeostasis and inflammation [40]. Thus, MAMs do not only serve as structural base for the localization of multiple molecular players, but also as a strategic platform for the decoding of signals of various nature.

A decade ago, Zhou and colleagues demonstrated that mROS can trigger NLRP3 inflammasome activation [41]. Under resting conditions, NLRP3 localizes to the cytosol, but once activated, NLRP3 relocates into mitochondria and MAMs together with its partner ASC in a ROS-dependent manner [41]. NLRP3 inflammasome activation is impaired by the inhibition of complex I or III of the mitochondrial respiratory chain or by the inhibition of the voltage-dependent anionic channel (VDAC), known to promote ROS generation [41]. Moreover, thioredoxin (TRX)-interacting protein (TXNIP), a protein involved in type 2 diabetes, redistributes to MAMs/mitochondria upon NLRP3 inflammasome activation in a ROS-dependent manner [42].

Furthermore, mROS, through the modulation of the NLRP3 inflammasome, can activate innate immunity [41,43].

In addition to alteration of inflammatory signaling pathways regulated by mROS, several studies have demonstrated that Ca^2+^-signaling is also important during NLRP3-mediated inflammation [44].

As in many other settings involving Ca^2+^ signaling, Ca^2+^ appears to be mobilized from the extracellular space as well as intracellular stores. Different hypotheses have been proposed for the role of Ca^2+^-signaling in NLRP3 activation [45], one of which is based on the role that Ca^2+^ sensing receptor (CASR) may have in the activation of NLRP3 by either increasing intracellular Ca^2+^ or decreasing cAMP [46]. Another model proposed that Ca^2+^ flux, through ER Ca^2+^ release channels, promotes mitochondrial Ca^2+^ overload and destabilization [47]. Supporting evidence by Misawa et al. reports that, during inflammasome formation, microtubules enhance the perinuclear migration of mitochondria, favoring the juxtaposition of all components for NLRP3 assembly between ER and the mitochondria [48]. These data suggest that the proximity between mitochondria and ER facilitates the transmission and propagation of Ca^2+^ signals between the two organelles, promoting the inflammatory response. 

No less important, mtDNA has been demonstrated to mediate the inflammatory response and to be important for caspase-1 activation. In response to LPS and ATP, inhibition of autophagic proteins leads to dysfunctional mitochondria and cytosolic translocation of mtDNA; in turn, cytosolic mtDNA contributes to downstream activation of caspase-1 [49]. Corroborating these results, Shimada and colleagues stated that oxidized mtDNA binds and activates the NLRP3 inflammasome [50].

Moreover, mitochondrial dynamics also are able to modulate NLRP3 inflammasome activation. In fact, knockdown of *Drp1* induces an increased mitochondrial elongation, which leads to NLRP3-dependent caspase-1 activation and IL-1β production in mouse bone marrow-derived macrophages [51]. In addition, chemical stimulators of mitochondrial fission, like carbonyl cyanide m-chlorophenyl hydrazine, clearly reduce NLRP3 inflammasome assembly and activation [51]. Recently, SESN2 (sestrin 2), known as stress-inducible protein, has been shown to induce mitophagy that removes the damaged mitochondria repressing prolonged NLRP3 inflammasome activation [52]. 

### 2.3. Mitochondrial Dynamics during Bacterial Infection

As already mentioned, mitochondria are the target of choice for viruses, but also several bacteria have been reported to manipulate mitochondria during infection. 

Mitochondrial Rho GTPases (Miro1 and Miro2) have been reported to regulate mitochondrial dynamics and in turn the mitochondria-dependent immune response during bacterial infection. During infection, a *Vibrio cholerae* Type 3 secretion system effector (VopE) localizes to the mitochondria, thanks to membrane potential, and acts as a specific GTPase-activating protein, which interferes with Miro1 and 2 [53]. Intriguingly, VopE increases MAVS aggregation and induces NF-kB signaling [53].

Furthermore, infection with *Brucella abortus* leads to altered mitochondrial energy production due to a metabolic shift to a Warburg-like state [54] and induces DRP1-independent mitochondrial fragmentation [55]. 

An example of how bacteria exploit the mitochondrial network to promote their own replication is infection with *Listeria monocytogenes*, which severely alters mitochondrial dynamics by causing mitochondrial network fragmentation and loss of MMP [56]

To favor its own replication, *Legionella pneumophila* interacts with mitochondria, inducing mitochondria fragmentation that finally leads to altered mitochondrial metabolism [57]. 

Another example of how bacteria create a favorable niche for their replication is *Chlamydia trachomatis,* an obligate intracellular human pathogen, which preserves mitochondrial integrity by inhibiting fragmentation and reducing DRP1 expression [58]. Furthermore, *C. trachomatis* initially takes advantage of host ATP mitochondrial production and then generates a sodium gradient to sustain its energetic demand [59].

Emerging evidence has demonstrated that mROS also facilitate antibacterial innate immune signaling and phagocyte bactericidal activity.

The SopB effector protein of *Salmonella typhimurium* suppresses mROS generation in response to infection to dampen the host immune response and to facilitate its establishment into the host cell [60]. SopB binds to cytosolic tumor necrosis factor receptor associated factor 6 (TRAF6), prevents its recruitment to mitochondria and inhibits apoptosis [60].

Infection of macrophages with methicillin-resistant *Staphylococcus aureus* (MRSA) induces mROS production that is IRE1α-dependent and triggers the generation of Parkin-dependent mitochondrial-derived vesicles (MDVs), which contribute to mitochondrial-peroxide accumulation in the bacteria-containing phagosome [61].

West et al. demonstrated that the mitochondrial adaptor protein ECSIT interacts with TRAF6 to upregulate mROS production in macrophages, which is essential for bactericidal activity following TLR1, TLR2 or TLR4 ligation [62]. 

Additionally, bacterial DNA is recognized by TLR9, a member of the highly conserved PRRs known as TLR. TLR9 recognizes unmethylated CpG dinucleotides, which are abundant in prokaryotic DNA and yet are rare in eukaryotic DNA. Nevertheless, mtDNA could be considered a ligand of TLR9 [63]. 

Taken together, these data pinpointed that, in addition to their well-established roles in the control of apoptosis and cellular metabolism, mitochondria are also intertwined in the innate immune response to cellular damage and appear to be pivotal hubs for innate immune signaling and the consequent generation of effector responses. 

## 3. Role of Mitochondria and Neuroinflammation in Neurodegenerative Diseases

Neurodegeneration is a pathological condition characterized by the progressive degeneration and loss of neurons and synapses in a particular area of the central nervous system (CNS). This degenerative process is based on a multifactorial mechanism, which involves genetics, aging, endogenous and environmental factors. Even if the basic molecular mechanisms beyond neurodegeneration are still not fully understood, neurodegenerative disorders (NDDs) can be grouped according to common pathogenic mechanisms: aberrant protein dynamics (misfolding, defective degradation, proteasomal dysfunction), oxidative stress and excessive ROS production, impaired bioenergetics with mitochondrial dysfunction and DNA damage, neutrophil dysfunction and neuroinflammatory processes [64]. 

In this review, we are going to focus on the impact of mitochondrial dysfunction and neuroinflammation in the development of NDDs from a clinical point of view.

Notably, local sterile inflammation has been shown to be finely linked with the development and the progression of different NDDs, such as Alzheimer’s disease (AD), Parkinson’s disease (PD), amyotrophic lateral sclerosis (ALS) and multiple sclerosis (MS) [65,66]. Neuroinflammation is commonly driven through the abnormal activation of brain immune cells, namely, microglia and astrocytes, by DAMP molecules released from damaged and necrotic cells [67,68]. Microglia cells represent the macrophage counterpart in the brain. They are responsible for the removal of damaged neurons and for monitoring pathogens. On the other hand, astrocytes account for the maintenance of brain structure and regulation of synapses and represent neuronal metabolic support [67]. Dysregulated activation of microglia and astrocytes results in persistent inflammasome activation, which, together with an increased level of DAMPs, leads to the establishment of low-grade chronic inflammation and thus to the development of age-related pathological processes [69]. Noticeably, neuroinflammation drives the increased secretion of cytokines and chemokines not only within the brain, but also systemically [70] and, in some cases, might lead to blood brain barrier disruption with the consequent infiltration of peripheral immune cells [67]. Accordingly, neuroinflammatory processes have been associated with metabolism alterations, such as obesity and type 2 diabetes [71,72]. At the molecular level, neuroinflammation is mainly triggered by redox status [73]: ROS are produced by microglia upon their activation by intrinsic or extrinsic factors (reviewed in [74]) and are released in the extracellular space. Uncontrolled ROS production might affect intracellular redox balance, thus inducing the expression of proinflammatory genes by acting as second messengers [75]. Consequently, abnormal activation of microglia leads to the release of reactive oxygen intermediates, proinflammatory cytokines, complement proteins and proteinases, driving a chronic inflammatory state responsible for triggering or maintaining neurodegenerative processes [76]. It is important to underline that redox-dependent pathways not only control inflammation but also are involved in different cellular functions, such as the regulation of metabolism, aging, proliferation, differentiation and apoptosis [77]. As already mentioned, the major players involved in these processes are mitochondria, as they are responsible for both generating ROS and responding to ROS-induced cellular changes [78]. Hence, mitochondrial dysfunction can be both the leading cause of neuroinflammation and can be induced by it. It is well known that mitochondria represent the cellular energy supply, and since the primary source of energy for the brain is glucose, alterations of neuronal glucose metabolism, mainly supported by mitochondria, lead to impaired cognitive functions [78]. The brain uses approximately 25% of the total glucose required by the body [79]; this is because neurons depend on OXPHOS to support their functions, like synaptic transmission and maintenance of neuronal potential [68,80,81]. 

Briefly, the OXPHOS pathway generates ATP using nicotinamide adenine dinucleotide (NADH) and flavin adenine dinucleotide (FADH2) produced by the TCA cycle. Therefore, due to the restricted ability of neurons to enhance glycolysis or to counteract oxidative stress, mitochondrial dysfunction leading to energy failure and oxidative damage is considered the basis of neuronal cell loss in neuroinflammation [82,83]. Consequently, prolonged inflammation, due to microglial inflammasome activation, oxidants and cytokine secretion (as H_2_O_2_, IL-1β and IL-18), combined with the alteration of neuronal cell metabolism are the triggering causes of neurodegeneration (Figure 2) [68]. Interestingly, aberrant activity of TCA enzymes, such as alpha-ketoglutarate dehydrogenase (KGDH), and pyruvate dehydrogenase (PDH) have been observed in the brain tissue of patients with AD [84,85]. 

More complex is the scenario in MS, where cellular metabolism varies according to the activity of the disease. Specifically, in active MS demyelinating lesions, characterized by the presence of activated microglia and macrophages, increased levels of PDH complex, malate dehydrogenase (MDH) and KGDH were detected compared to normal white matter, indicating that glycolytic and TCA cycle pathways were increased [86]. On the contrary, in demyelinated axons, namely, inside the inactive lesion (plaque), KGDH activity was reduced, thus correlating with terminal axonal damage [86]. These findings suggest that the TCA cycle is definitely altered in NDDs, but upregulation or downregulation of its enzymes might change according to the type of disease and the type of cells involved. 

Oxidative stress and mitochondrial metabolism are finely linked in NDDs: myeloperoxidase activity has been shown to be upregulated in the microglia of patients with AD and the brains of patients with MS [87,88], and myeloperoxidase products, namely, hypochlorous acid and chloramine, can inhibit KGDH activity, thus revealing its sensitivity to inflammatory ROS.

Administration of glutathione, a well-known antioxidant, was able to restore KGDH activity after peroxynitrite treatment in neuroblastoma cell cultures [89]. Taken together, these results suggest that during neuroinflammation, TCA cycle enzyme activity is affected, indicating the possible involvement of inflammatory mediators (as ROS) in altering mitochondrial metabolism. 

This hypothesis is supported by studies conducted on different cell types and tissues, such as cardiomyocytes, fibroblasts, skeletal muscle and liver, in which inflammatory cytokines have been shown to influence TCA cycle components, mainly reducing PDH activity and therefore acetyl-CoA production via glycolysis. 

Interestingly, genes encoding ETC complexes have been shown to be downregulated in patients presenting mild cognitive impairment, AD or MS. In AD patients, upregulation of immune genes was reported, confirming the correlation between impaired cell metabolism and inflammation [90]. In both AD and MS, microglia are persistently activated with the consequent release of inflammatory mediators, such as ROS, cytokines and chemokines, which leads to OXPHOS impairment in neurons and other glial cells [83]. Several studies have shown how TNFα bursts, either induced by lipopolysaccharide (LPS) or by direct TNFα administration, affect OXPHOS complexes in the liver, primary neuronal cell culture and cardiac muscle [91,92,93,94,95]. At the same time, TNFα inhibition restored complex III and ATP synthase activity [96]. TNFα is a pleiotropic cytokine that is able to induce cell death by binding to the p55 receptor (TNF receptor 1). Interestingly, it has been shown that in the neurodegenerative context, TNFα induces neuronal cell death by silencing survival signals (SOSS), such as phosphatidylinositol 3’ kinase, indicating SOSS inhibition as a possible treatment for neurodegenerative disorders [97]. Noticeably, TNFα treatment promotes mitophagy in neuro-blastoma cells [98]. TNFα exerts its proinflammatory role also by reducing peroxisome proliferator-activated receptor (PPAR)-γ coactivator 1α (PGC-1α) expression [99] and by increasing mitochondrial fragmentation acting on OPA1 isoform balance [100], even if these effects have not been investigated in nervous tissue yet. 

However, PGC-1α has been found to be downregulated in NDD neurons, and IL-1β has been shown to induce mitochondrial fragmentation and to impair the respiration rate in astrocytes, underlying the role of proinflammatory cytokines in regulating mitochondria dynamics in the CNS [101]. 

Damaged mitochondria trigger the process of mitochondrial membrane permeabilization, known to be the starting point of both apoptosis and necrosis. Apoptosis is essential for nervous system development, whereas adult neurons are resistant to this form of cell death [102]. Recently, different studies have shown how necroptosis, a form of regulated necrotic cell death induced by TNFα, is highly activated in NDDs [102,103,104]. Briefly, necroptosis is a form of programed cell death coordinated by receptor-interacting kinases 1 (RIPK1) and RIPK3 and mixed lineage kinase domain-like protein (MLKL) under caspase-8 deficient conditions. Necroptosis can be stimulated by TNF, other members of the TNF death ligand family, interferons, Toll-like receptor signaling and viral infection [105]. In MS patients, the RIPK1-RIPK3-MLKL pathway is activated, and experiments in animal models showed that oligodendrocytes necroptosis could be blocked by RIPK1 inhibition [66,106].

Lastly, as previously mentioned, accumulation of damaged mitochondria can activate NLRP3 inflammasome-dependent inflammation in microglia. Moreover, damaged neurons are responsible for releasing DAMPs, such as mtDNA, in the extracellular environment, eliciting local inflammation [49] by increasing inflammasome activation, and thus IL-1β secretion, and also binding to microglial toll-like receptor-9, inducing TNFα and nitric oxide (NO) production [107]. 

In conclusion, neuroinflammation affects many mitochondrial processes such as TCA, OXPHOS, fusion and fission, membrane permeabilization, and mitophagy and might also induce the accumulation of mtDNA mutations, impairing energy production and thus distressing cognitive ability, even if a direct correlation between alterations of these cellular functions and patient clinical outcomes is still under investigation. Moreover, extracellular release of mitochondrial components (mtDNA or proteins) acts as an inflammatory boost, leading to a vicious inflammatory cycle. Therefore, from a therapeutic point of view, suppression of microglia-mediated inflammation can be considered an important curative strategy for NDDs. Accordingly, different nonsteroidal anti-inflammatory drugs (NSAID) repress microglial activation, targeting cyclooxygenase (COX) or PPAR-γ and exerting neuroprotective effects in the CNS [76,108]. Several studies suggested inhibition of Rho kinase (ROCK) signaling, shown to be involved in mitochondrial fission, as a promising treatment option for NDD [109]. Interestingly, a recent study revealed that curcumin, the main curcuminoid isolated from *Curcuma longa* (turmeric), significantly reduced TNFα, prostaglandin E2 (PGE2), and NO secretion in in vitro lipoteichoic acid-activated microglial cells. Curcumin also inhibited NO synthases (iNOS) and COX-2 expression [110]. Similarly, the ent-kauranoid diterpenoid glaucocalyxin B, isolated from the aerial parts of *Rabdosia japonica*, decreased NO, TNFα, IL-1β, COX-2 and iNOS in LPS-activated microglia cells [111]. These plant derivatives might represent a promising approach in targeting neuroinflammation, and different studies are aiming to define their optimal bioavailability, even if their use in clinic has yet to be defined [112,113]. Another therapeutic approach to reduce neuroinflammation is blocking necroptotic pathways with synthetic inhibitors (RIP1 inhibitors, RIP3 inhibitors, MLKL inhibitors) to mitigate NDD progression [114]. Drug treatments aimed at manipulating inflammasome assembly therapeutically are currently used in clinic [115], but, despite their promising efficacy, they still do not resolve the disease. The low efficacy of actual therapies in treating NDDs might be due to the complexity of neurodegenerative processes, which involve neuroinflammation but also have roots in genetic predisposition and environmental factors. Nevertheless, reduction in inflammation ameliorates disease progression, therefore representing a fundamental complementary therapy. Since neuroinflammation is one of the triggering causes of NDDs, therapies aimed at reducing the inflammatory process result in a reduction in disease progression, therefore representing a fundamental complementary therapy.

## 4. Inflammation-Related Mitochondrial Dysfunction in Cancer: A Negative Loop 

In 1893, Rudolf Virchow was the first to hypothesize that chronic inflammation and tumorigenesis were connected [116]. This assumption was deduced from the presence of leukocytes in cancerous lesions and from the idea that inflammation caused by injuries and irritants can contribute to cell proliferation.

Further studies indicated that inflammation represents a potential risk factor for tumorigenesis; indeed, some inflammatory conditions are related to malignant transformation. Some examples of this correlation can be found in *Helicobacter pylori*-caused gastritis and gastric cancer, gut pathogens involved in inflammatory bowel diseases and colon rectal cancer, or human Papilloma virus’ cervicitis and cervical cancer, but also in chemical/physical irritants such as asbestos fibers that lead to asbestosis and mesothelioma or UV rays that cause sunburns and melanoma, as well as tobacco and alcohol, which cause bronchitis or pancreatitis, respectively, leading to lung cancer and pancreatic/liver cancer [117].

Recent studies demonstrated that approximately 25% of tumor malignancies are related to chronic inflammation and pathogen infection [118]. As a matter of fact, cancer-related inflammation is the 7th hallmark in tumor development [119]. 

Hanahan and Weinberg reviewed that cancer-related chronic inflammation drives unlimited proliferative potential, independence from growth factors, enhanced angiogenesis, metastasis, escape from apoptosis and resistance to growth inhibition [120].

Further evidence demonstrated that the tumor microenvironment, rich in inflammatory cells, fosters proliferation, survival and migration of tumors. Moreover, cancer cells appropriate some of the signaling molecules of innate immunity to promote invasiveness and proliferation [121].

To summarize, deficient pathogen eradication or recurring injuries, as well as prolonged inflammatory signaling, support cancer development, so that tumors can be considered a failed wound-healing process [117]. 

It is largely known that mitochondria regulate the metabolic needs of cells and decide between life and death upon different conditions. Recently, it has been demonstrated that MOM permeabilization exposes cells to considerable proinflammatory effects; among these, the release of mtDNA from the mitochondria leads to the IFN-I response and NF-kB proinflammatory signaling via the IAP-regulated mechanism [122]. 

Otto Warburg, in the 1930s, was the first to link mitochondrial dysfunctions to cancer. Indeed, he observed that tumor cells display an increased rate of aerobic glycolysis; from this evidence, he speculated that the production of ATP via glycolysis instead of oxidative phosphorylation might be explained by mitochondrial respiratory capacity impairment.

Moreover, several studies have proposed that altered cancer cell metabolism, defined as acidic and/or ischemic, that enables survival in a hostile environment represents a strategy of evasion from the attack of immune system cells, thus reinforcing cancer stem cell resistance [123].

In past decades, a large amount of data has been collected on the possible mitochondrial defects that can lead to cancer, such as alteration in the activity and expression of the mitochondrial respiration complex (MRC) and mtDNA mutations [124]. 

Of note, dysfunction in MRC Complex I is associated with Hürtle cell tumors of the thyroid [125], and a decrease in Complex III activity can be linked to breast cancer, [126] and Hürtle cell tumors [127,128]. Moreover, decreased activity of Complex II, III, and IV is associated with the aggressiveness of renal cell tumors [129].

Regarding mtDNA, many lines of evidence show correlation between mutations and ovarian, kidney, thyroid, liver, gastric, colon, lung, head and neck, brain, breast, and bladder cancers and leukemia [130].

Even though the link between key mitochondrial players and cancer is quite well established, the connection between MRC dysfunction and mtDNA mutations in cancer and inflammation remains to be fully elucidated. In a vicious cycle, the impairment of respiratory chain proteins and mitochondrial genes causes an increase in mitochondrial ROS production, as these species are the cause of damage. Among mitochondrial dysfunctions, it is well established that ROS are important signaling molecules that participate in cell migration and invasiveness, proliferation, migration and gene transcription. In addition, ROS levels increase upon mitochondrial malfunction, as tumor cells usually have higher ROS production compared to normal cells [2]. 

Intriguingly, a connection between ROS and hypoxia-inducible factor (HIF) has been reported. HIF is a crucial transcription factor for cancer metabolism and metastasis, and it is activated upon hypoxic conditions. ROS, though, are able to stabilize HIF even under a normal oxygen concentration, thus promoting tumorigenesis [131]. 

Moreover, lack of succinate dehydrogenase (Complex I) and fumarate hydratase in tumors is related to difficulty in HIF degradation [132].

Additionally, a ROS increase results in activation of the NLRP3 inflammasome and release of proinflammatory cytokines, such as IL-1β, that suppress immune-surveillance, enabling tumor progression [9].

Thus, chronic inflammation diseases are characterized by an overproduction of free radicals, often concomitant with a decreased capacity to scavenge them [133]. When ROS reach a dangerous level, cancer cells increase mitochondrial production of NADPH to diminish the effect of ROS to evade apoptosis [134].

ROS represent a threat to all biological macromolecules in the cell; in particular, they cause oxidation to the fatty acids of plasma membranes, proteins and genes, causing mutations and cancer-related alterations [135]. 

Upon the activation of innate immune cells, there is secretion of proinflammatory soluble molecules, cytokines and chemokines that induce ROS production (Figure 3). In a chronic inflammation context, ROS production from immune system cells can drive to cell damage or hyperplasia [136]; for instance, IL-1, IL-6, TNFα and IFN-γ induce ROS production in nonphagocytic cells as well. 

Indeed, in the inflammatory process, TNFα activates the transcription factor NF-kB, which in turn induces the expression of genes involved in cell proliferation, carcinogenesis and blockage of apoptosis, as well as the production of proinflammatory cytokines to potentiate the response. In this context, ROS have a dual role: (i) they can be considered a potential threat when overproduced by mitochondria in tumorigenesis, as well as enhancers of cancer proliferation through the activation of proliferation pathways via the innate immune system, together with the inactivation immune surveillance; and (ii) they can sustain a chronic innate immune response.

As stated, nucleic acids can be severely damaged by free radicals, especially mtDNA, since they lack the protection of histone proteins. In a dangerous loop reaction, ROS induce mutations in mtDNA, leading to dysfunction in the production of proteins of the respiratory chain, thus enhancing the production of ROS [137]. An alternative study conducted by Trifunovic and collaborators reported that mtDNA mutator mice, an aging model, did not show either an increase in ROS production or an increased sensitivity to oxidative stress-induced cell death despite the accumulation of mtDNA mutations in a linear manner, even in the presence of severe respiratory chain impairment [138]. Even though this study gives an alternative point of view on mtDNA mutation-dependent ROS production, it would be interesting to understand which mitochondrial genes are more susceptible to mutation in order to define whether there can be a direct correlation between mtDNA mutations and ROS production.

Although mtDNA is the most harmed macromolecule, it fulfills a double role in cancer-related inflammation. Recently, several pieces of evidence showed a horizontal transfer of mitochondria between tumor cells and surrounding nontumor cells, so it has been hypothesized that this transfer is a strategy that cancer cells adopt to counteract mitochondrial dysfunction and satisfy high metabolic requirements [139]. Tan and collaborators have demonstrated that when there is a lack of mtDNA, there is a decrease in tumor growth, so tumor cells start to acquire mtDNA molecules from healthy surrounding cells, restoring their metabolic functions and their tumorigenic potential, most likely with the aid of the mitochondria themselves [140].

To date, it remains undefined how this phenomenon is triggered and how healthy cells can be persuaded. Some in vitro evidence has shown that solid tumor cell lines are able to acquire healthy mitochondria from MSCs (mesenchymal stem cells) and endothelial cells, possibly suggesting the participation of the circulatory system in this process. For a complete overview on the topic and possible strategies for mtDNA transfer, we suggest the review by Berridge and collaborators [141].

It has been reported that cancer progression is promoted by mtDNA molecules in the surrounding microenvironment and circulating system, which in turn regulates the production of proinflammatory cytokines suppressing antitumor effects exerted by the immune system. Dendritic cells, through STING signaling, recognize circulating mtDNA activating lymphocytes to trigger tumor suppression, but the inappropriate sensing of mtDNA results in a deficiency of cancer cell clearance by those cells [142]. Furthermore, mtDNA transfer from cancer cells to the microenvironment enriched in immune cells, such as monocytes and natural killer cells, leads to their dysfunction and apoptosis, thus reinforcing tumor progression; for a complete overview refer to Liu et al. [143].

Thus, mutated mtDNA can either be the victim or the inducer of ROS production (from both mitochondria and the innate immune system), possibly representing a risk or an advantage for cancer proliferation. However, circulating wild-type mtDNA transferred by surrounding cells can provide mutations in cancer cells, resulting in evasion of death.

Mitochondrial dysregulation does not involve only free radical production or mtDNA mutation and/or transfer; indeed, the literature is full of evidence showing the involvement of Ca^2+^ homeostasis alterations in cancer. This occurs also at MAMs, which provide the allocation of important oncogenes or oncosuppressors, such as p53, PML, PTEN, kRAS, Bcl-2, and Bcl-Xl [144,145,146,147], all of which are known to reduce mitochondrial Ca^2+^ uptake, thus blocking apoptosis in cancer models. 

As stated, cancer cells are marked by a defective metabolism, so therapies targeting glycolysis, pyruvate oxidation and glutamine metabolism would have a protective effect from chronic inflammation as well [148].

Targeting metabolic traits can be tricky though, since immune and cancer cells have some metabolic differences, and tumor metabolic pathways must be shut down in order to trigger the apoptotic event. Moreover, cancer and immune infiltrating cells reside together in a microenvironment where signaling molecules influence the metabolic choices of both [149]. Tumor-associated macrophages (TAMs) are the best example of metabolic reprogramming exerted by cancer cells because this promotes the increase in protumoral factor production, fostering protumorigenic inflammation conditions that favor the proliferation and metastasis of cancer. TAMs are essentially “educated” by the tumor to let them deliver essential molecules to sustain their proliferation [150].

Together with TAMs, in infiltrating tumor cells it is possible to find several populations of T lymphocytes, particularly T helper 17 cells (Th17) [151] inhibited by the tumor itself. Indeed, it has been proposed that their suppression corresponds to the promotion of regulatory suppressive Treg cells, a population associated with tumorigenesis boost [152]. The impairment of mitochondrial metabolism and failure of immunosurveillance create the conditions for immune-associated diseases, and it is clear by now that cancer is one of these [152] (Figure 3).

The relationship between inflammation and cancer is thus established as a vicious cycle in the mitochondria, where infiltrating immune cells are recruited upon proinflammatory signaling aiming at the eradication of cancer, which is recognized as an inflammatory disease; at the same time, the tumor mass can take advantage of the inflammatory microenvironment to sustain its metabolism. ROS have a crucial role throughout the whole process. Indeed, ROS produced by the immune response aid tumor growth, while on the other hand, cancer cells with impaired mitochondria increase ROS production as well, leading to potential harm that can be evaded through horizontal mtDNA transfer and the upregulation of detoxifying systems.

Recent research showed that the Warburg effect is favorable to tumor cells because it upregulates antioxidant enzymes such as glutathione reductase, superoxide dismutase, catalase, peroxiredoxin, and thioredoxin to diminish the overproduction of ROS [153] that harm mitochondrial macromolecules (i.e., mtDNA and protein), as previously stated. 

Another enzyme that plays an important role in this process is the glycolytic enzyme pyruvate kinase M2; cancer cells express the embryonic isoform (PKM2) [154]. The increase in ROS inhibits PKM2 leading to a switch from glycolysis to pentose phosphate metabolism to produce reducing equivalents (NADPH), which are essential to maintain antioxidant molecule activity and necessary for ROS detoxification [155]. In this way, the regulation of PKM2 gives cancer the capability to resist the increase in mtROS production [153].

Fortunately, much effort has been expended in developing therapies against this vicious cycle, all of them targeting apoptosis induction; these therapies are grouped in a category of molecules called mitocans (mitochondrial targeted anticancer drugs).

A group of them includes vitamin E analogs that enhance ROS production in the mitochondria, favoring the induction of apoptosis specifically in cancer cells because they are under greater oxidative stress compared to normal cells [124,156]. These mitocans can also be designed for specific mitochondrial targets, for example components of the OMM, inter-membrane space, *cristae* or the matrix. Specifically, these targets, such as hexokinase (HK), VDAC and adenine nucleotide translocase (ANT), can be involved in the production and shuttling of ADP/ATP; mitocans against these proteins disrupt the main metabolic processes and the antioxidative capability of the tumor [157].

There is also clinical approval for BH3 mimetics that target antiapoptotic proteins of the Bcl-2 family [158,159] for chronic lymphocytic leukemia [160].

ETC is another powerful target for specific inhibitors since its disruption impairs oxidative phosphorylation [161]. 

Moreover, a novel mitochondrial inhibitor named CPI-613 has been approved for a phase I clinical study with pancreatic cancer patients. This compound blocks two enzymes, pyruvate dehydrogenase and a-ketoglutarate dehydrogenase [162,163], and was administered in combination with modified FOLFIRINOX; the combination has given improved outcomes [164,165].

Nevertheless, other molecules are undergoing optimization processes, and these are compounds that can hinder mitochondrial transmembrane potential, such as lipophilic cations [152]; the Krebs cycle, consequently affecting ECT and OXPHOS [166]; inhibitors against the conversion of pyruvate to AcCoA, hindering the Krebs cycle [167]; or inhibitors that interfere with DNA polymerase, which influences mtDNA transcription, with consequent effects on the mitochondrial protein pool [168,169].

All of this evidence supports mitochondria as new pharmacological targets for patients with cancer. Obviously, more work is necessary to shed light on how mitochondrial dysfunction modulates the inflammatory response associated with tumor development. 

## 5. Conclusions

Besides their ancestral function as the powerhouse of eukaryotic cells, mitochondria have gained attention as principal intracellular signaling platforms associated with innate immunity and inflammation. It is now clear that mitochondrial derangement can be considered a crucial pathogenic mechanism of several diseases characterized by chronic inflammation, including neurodegenerative diseases, cancer, rheumatoid and metabolic disorders.

Mitochondrial defects that are able to trigger inflammation are not only limited to metabolic variation or imbalance of the physiological mechanisms that control shape and number but could also be extended to other mitochondrial pathways. As an example, chronic stresses capable of increasing mitochondrial Ca^2+^ entry induce sustained inflammation, and MCU-mediated Ca^2+^ overload is essential for triggering NLRP3-mediated inflammation in patients with cystic fibrosis during infection with *Pseudomonas aeruginosa* [170]. Together with the role of MCU in the regulation of decreasing IFN-β levels induced by viral infection discussed above [27], these observations suggest that the MCU complex could be identified as a potential target in the treatment of inflammation-associated diseases. Future work is required to evaluate if other components of the mitochondrial Ca^2+^ machinery, for example the Ca^2+^ efflux system [171], are involved in the control of inflammation.

Nevertheless, other important issues remain to be clarified. As the interface between mitochondria and MAMs has an important role in controlling the inflammation response [172], it is tempting to speculate that other mitochondrial molecules can regulate MAVS signaling and/or be critical for RLR signaling. Do the mitochondria also translate other innate immunity signaling? 

Therapies that target mitochondria-NLRP3 inflammasome activation or mitochondrial dysfunction affecting immune cell function may hold great promise in the treatment of diverse inflammation-mediated diseases. Therefore, answers to these questions may provide new pharmacological targets for the treatment of acute and chronic pathological and inflammatory disorders.

## Figures and Tables

**Figure 1 jcm-09-00740-f001:**
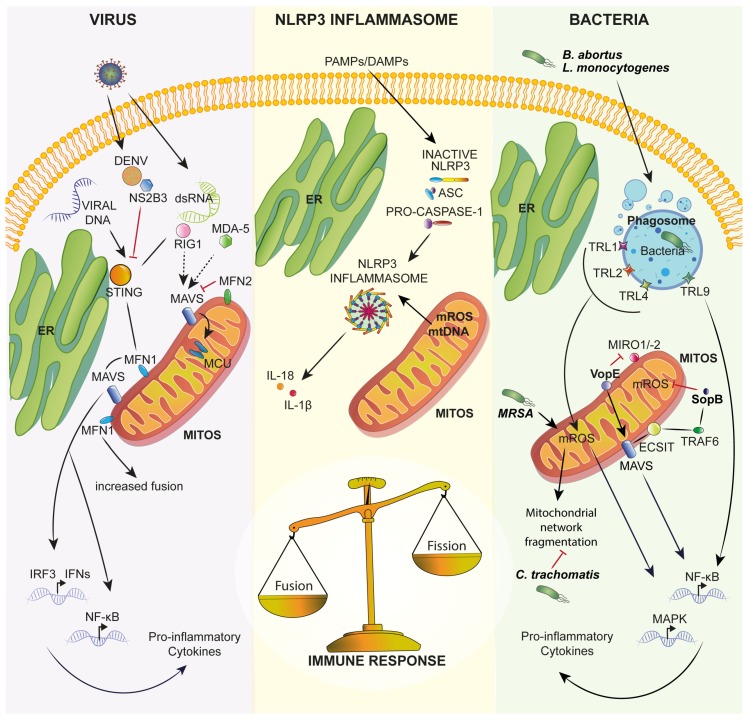
Schematic representation of the role of mitochondria in innate immunity. Viral infection, exposure to PAMPs/DAMPs or exposure to bacteria can activate the immune response, altering mitochondrial dynamics and functions. Upon viral infection, MAVS plays a key role in activation of the innate immune response, activating NF-kB and IRF-3 (interferon regulatory factor 3) signaling and inducing proinflammatory cytokine and type I interferon release. Moreover, mitochondria are a key source of DAMPs that are able to activate the NLRP3 inflammasome, leading to proinflammatory cytokine release such as IL-1β and IL-18. Several bacteria manipulate mitochondria during infection. See text for further details. DAMPs: damage-associated molecular patterns; DENV: dengue virus; ER: endoplasmic reticulum; NLRP3: NOD-, LRR- and pyrin domain-containing 3; MCU: mitochondrial calcium uniporter; MDA-5: melanoma differentiation-associated gene 5; MFN: mitofusin; mROS: mitochondrial reactive oxygen species; PAMPs: pathogen-associated molecular patterns; RIG-1: retinoic acid-inducible gene I; TLRs: Toll-like receptors; TRAF6: tumor necrosis factor receptor associated factor 6.

**Figure 2 jcm-09-00740-f002:**
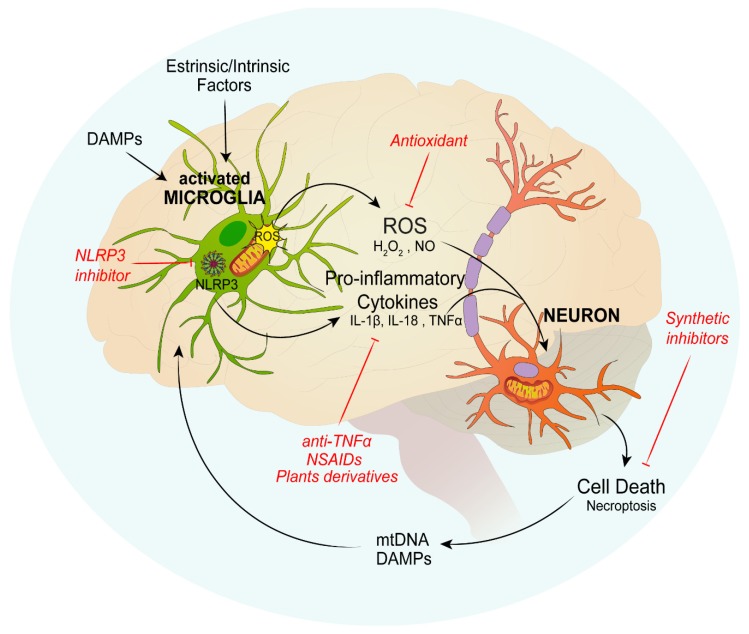
Involvement of mitochondrial dysfunction and neuroinflammation in the development of neurodegenerative diseases. DAMPs or extrinsic/intrinsic factors activate microglia that lead to ROS production and proinflammatory cytokines release that in turn alter neuronal functions and induce cell death. This creates a vicious cycle that favors mtDNA and DAMP production, activating the NLRP3 inflammasome. Selective NLRP3 inhibitors, antioxidants, specific inhibitors that block the necroptotic pathway, anti-TNFα, NSAIDs and plant derivatives can prevent this mechanism. DAMPs: damage-associated molecular patterns; NLRP3: NOD-, LRR- and pyrin domain-containing 3; NSAID: nonsteroidal anti-inflammatory drugs; mtDNA: mitochondrial DNA; ROS: reactive oxygen species.

**Figure 3 jcm-09-00740-f003:**
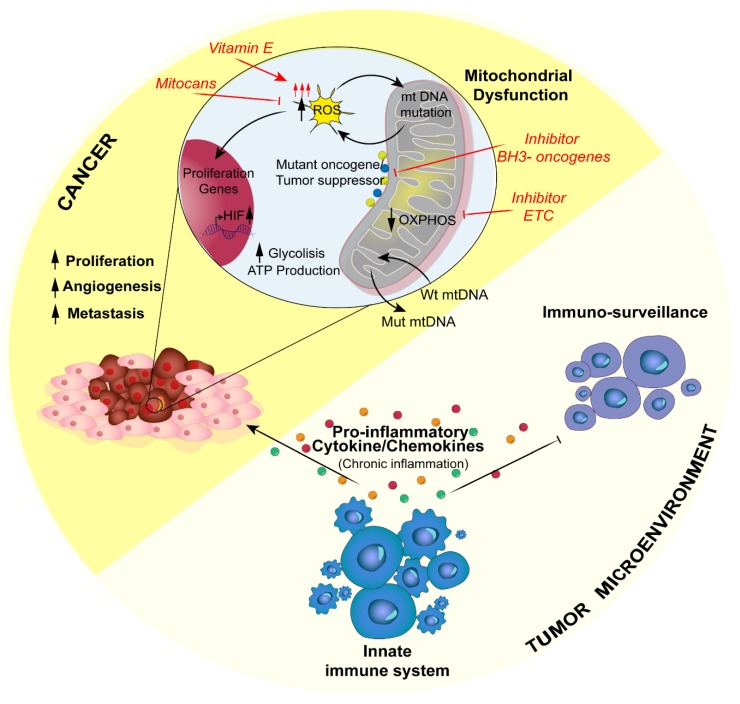
Schematic representation of how mitochondrial dysfunctions, which alter the inflammatory response, can promote cancer. In cancer, mitochondrial dysfunctions lead to increased ROS production, which can have a dual role that is either dangerous or prosurvival. On the one hand, mitochondrial ROS (mROS) can accumulate DNA mutations; on the other hand, mROS can increase cytokine inflammatory release from the innate immune system. A strategy that cancer cells can adopt to survive the increased rate of mutations is represented by the horizontal transfer of wild-type mtDNA molecules from surrounding healthy cells. The inflammatory response by the innate immune system can sustain cell tumor growth instead of counteracting it, altering the immune-surveillance and STING pathways (see text for further details). The main clinical strategy could be the selective inhibition of mROS production. ETC: electron transport chain; MOMP: mitochondrial outer membrane permeabilization; mtDNA: mitochondrial DNA; OXPHOS: mitochondrial oxidative phosphorylation; PKM2: pyruvate kinase M2; ROS: reactive oxygen species.

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
