# Peer review of "The Role of Mitochondria in Inflammation: From Cancer to Neurodegenerative Disorders"

_jcm, 2020, doi:10.3390/jcm9030740_

Round 1

Reviewer 1 Report

The review of Missiroli et al. is claimed to explore the current knowledge on interplay between mitochondria and inflammatory diseases. However, my major concern is that it is not a thorough review. The paper content would be more suitable as a discussion section for some original paper. It is hard to say who might benefit from reading it. It has little bit of this and little bit of that and it is simultaneously far too general to be of use for basic science researchers and too “biochemical” for clinicians. The clinical context, e.g. association of mitochondrial dysfunction with prognosis, advancement, therapy, etc. is almost non-existent and limited to cancer. The title is misleading - out of numerous inflammatory diseases, the authors discuss only two, for which inflammation is merely one of components in their multifactorial pathogenesis. What was the rationale for selecting neurodegenerative diseases AND cancer? It is expected that the review should be an overview of resent original research; here, however, the authors far too frequently cite other reviews instead of original papers. Abstract and introduction give an impression that the authors run in circles while talking about a relationship between mitochondria and chronic inflammation. In addition, there are factual errors throughout the text, such as:

- line 35: Ca2+ is calcium ion while calcium is denoted by Ca.

- line 19-22: the second statement does not follow from the first as suggested by “thus”. Persistent and/or inappropriately regulated inflammatory response is defining chronic inflammation (as mentioned in the first sentence) contrary to acute inflammation (mentioned in the second sentence);  

- line 89-90: the statement implies that metabolic quiescence equals catabolism and metabolic activity equals anabolism, which is not true – a cell can be metabolically less or more active and the activity can be shifted towards anabolic or catabolic processes;

- line 105: the statement implies that ATP synthesis is synonymous with ROS generation, which is not true - vide substrate level phosphorylation;

- line 106: oxidative stress, by definition, is an imbalance, a state not a process;

- line 144: I have doubts whether ROS can be included as DAMPS, oxidatively modified macromolecules – yes, but ROS?

- lines 320 and 342: pyruvate dehydrogenase is not a TCA enzyme;

- lines 550-551: where is FADH2 (and not FADH, by the way) produced in pentose phosphate pathway? How is it used for ROS detoxification?

Minor: numerous grammatical and editorial errors, e.g.:

different font types in the abstract; inflammatory diseases (line 27); “organismal” sounds a bit strange (line 99, 101); remove “-“ (line 88); as a physiological signals (line 107); receptors (line 125); discuss sth, without preposition (line 135); mediators (line 140); “are in common with several microorganisms” should rather be replaced with “are common for several microorganisms” (line 144); IFNβ has already been mentioned in line 152 (line 164); genes should be written in italics (line 255); the inclusion starting from “like...” ends with “...hydrazone” and should be ended with comma (line 258); why capitalize first word in full names of MS and ALS (line 282)?; redox what? (line 296).

Author Response

The review of Missiroli et al. is claimed to explore the current knowledge on interplay between mitochondria and inflammatory diseases. However, my major concern is that it is not a thorough review. The paper content would be more suitable as a discussion section for some original paper. It is hard to say who might benefit from reading it. It has little bit of this and little bit of that and it is simultaneously far too general to be of use for basic science researchers and too “biochemical” for clinicians.

R: We thank the reviewer for the constructive comments and valuable suggestions which strengthen our manuscript. We added appropriate references in each part of the manuscript. We re-organized the text trying to adress the topics we have dealt with.

The clinical context, e.g. association of mitochondrial dysfunction with prognosis, advancement, therapy, etc. is almost non-existent and limited to cancer.

R: Unfortunately, a direct association between mitochondria defect and disease prognosis (patients’ outcome) in both NDDs and cancer is still under investigation. What is known is that inflammation in both the cited pathologies is responsible for disease progression. The aim of this review is to connect these two aspects highlighting from a clinical point of view that targeting inflammation, also by acting on mitochondria, is a promising therapeutic approach. Sentences to clarify this point have been added in the text (lines 468-469, 483, 492-494).

The title is misleading - out of numerous inflammatory diseases, the authors discuss only two, for which inflammation is merely one of components in their multifactorial pathogenesis. What was the rationale for selecting neurodegenerative diseases AND cancer?

R: We agree with the reviewer that the chosen title is misleading, therefore we changed it to “Role of mitochondria in inflammation: from cancer to neurodegenerative disorders ”. The aim of the review is to elucidate how inflammation can be an important factor in disease progression, and what is the role of mitochondria in the inflammatory processes. We chose NDDs and cancer because in both these pathologies it has been demonstrated that, besides being only one of the multifactorial pathogenesis components, chronic inflammation relates with disease progression. Moreover, our recent works are related to these pathologies so we decided to focus our attention on what we actually work on.

It is expected that the review should be an overview of resent original research; here, however, the authors far too frequently cite other reviews instead of original papers.

R: We agree with the reviewer and we added appropriate references in each part of the manuscript.

Abstract and introduction give an impression that the authors run in circles while talking about a relationship between mitochondria and chronic inflammation.

R: We have re-organized both abstract and introduction in order to address the focus of the review. We hope that the reviewer could appreciate the current version of the manuscript.

In addition, there are factual errors throughout the text, such as:

- line 35: Ca2+ is calcium ion while calcium is denoted by Ca.

R: The reviewer is right and we apologize for the mistake. Accordingly, the abbreviation has been corrected (line 36).

- line 19-22: the second statement does not follow from the first as suggested by “thus”. Persistent and/or inappropriately regulated inflammatory response is defining chronic inflammation (as mentioned in the first sentence) contrary to acute inflammation (mentioned in the second sentence);  

R: The reviewer is right. We now re-organized the abstract, hoping that the current version will be more exhaustive (lines 18-27).

- line 89-90: the statement implies that metabolic quiescence equals catabolism and metabolic activity equals anabolism, which is not true – a cell can be metabolically less or more active and the activity can be shifted towards anabolic or catabolic processes;

R: The sentence has been reformulated (lines 115-118).

- line 105: the statement implies that ATP synthesis is synonymous with ROS generation, which is not true - vide substrate level phosphorylation;

R: The sentence has been reformulated (line 108).

- line 106: oxidative stress, by definition, is an imbalance, a state not a process;

  1. The sentence has been corrected (line 109).

- line 144: I have doubts whether ROS can be included as DAMPS, oxidatively modified macromolecules – yes, but ROS?

R: The reviewer is right and we apologize for the mistake. ROS cannot be included as DAMPs. The sentence has been corrected (line 160).

- lines 320 and 342: pyruvate dehydrogenase is not a TCA enzyme;

R: We agree with the reviewer, PDH was mistakenly included in TCA enzymes due to its involvement in the production of Acetyl-CoA. The sentence has been corrected (lines 404-406).

The meaning of this sentence is that the aberrant function of PDH results in reduced production of Acetyl-CoA from glycolysis, therefore TCA cycle will be affected. A sentence has been added to clarify the concept (lines 424-427)

- lines 550-551: where is FADH2 (and not FADH, by the way) produced in pentose phosphate pathway? How is it used for ROS detoxification?

R: Now line 654, we agree with the reviewer and we apologize for the mistake. FADH2 and FADH were mistakenly included. The sentences have been corrected.

Minor: numerous grammatical and editorial errors, e.g.:

different font types in the abstract; inflammatory diseases (line 27); “organismal” sounds a bit strange (line 99, 101); remove “-“ (line 88); as a physiological signals (line 107); receptors (line 125); discuss sth, without preposition (line 135); mediators (line 140); “are in common with several microorganisms” should rather be replaced with “are common for several microorganisms” (line 144); IFNβ has already been mentioned in line 152 (line 164); genes should be written in italics (line 255); the inclusion starting from “like...” ends with “...hydrazone” and should be ended with comma (line 258); why capitalize first word in full names of MS and ALS (line 282)?; redox what? (line 296).

R: We thank the reviewer for carefully reading our manuscript. We have checked the english grammar and we have corrected the numerous errors throughout the manuscript.

The review has been also edited by a professional English service.

Reviewer 2 Report

This review focuses on the emerging role of mitochondria in inflammation with connections to innate immunity and neurodegenerative disease and cancer. Thank you for the review, I think it is well organised and descriptive. Please see me few points for small areas of improvement.

1) There are very few errors, but please have someone check for english grammar, this mainly pertains to singular/plural rules.

2) Line 97: Can the authors clarify what they mean by "...mitochondria are tolerated by immunity...". Do they mean that these organelles are not attacked by the immune system?

3) Lines 118-128: It is suggested that the authors strengthen their argument supporting the role of mtROS activating the inflammasome. Is this due to evidence that mitochondria are the main producers of ROS. In the same line, is it mtROS and/or mtDNA and/or pro-inflammatory cytokines that are activating the inflammasome. Is it the release of any of these mitochondrial components (or just one) that is activating the inflammasome.

4) Line 135-136 "...it is a great promise for treatment of different inflammatory diseases?" is not a question.

5) Regarding Figure 1, is viral RNA only targeting/influencing mitochondria (via MAVS) or this is a simplified figure? This should be clarified in the figure legend and or supported text. Also in regards to this figure, MAVS activation appears to be inducing the release of Nf-kB etc, is this release due to loss of membrane potential and the concomitant initiation of apoptosis?

6) Lines 226-228, the authors state that under "...resting conditions NLRP3 is in the cytosol but once activated it migrates to the mitochondria linked by MAMs...". Above, however, the authors write of the importance of the activation of the inflammasome due to mtROS, what then is "driving" the movement from the cytosol to the interaction with mitochondria, can it still be mtROS even if at rest/unstressed it is further away from the source of ROS? For me, this needs to be clarified.

7) Line 488, although there is no evidence suggesting that mtDNA interacts with histone proteins, mtDNA exists in nucleoids and interact with TFAM, polymerase gamma, twinkle, single stranded binding protein. This should be made more clear to the audience. In this line, it is recommended that the authors make more clear the evidence presented linking mtDNA damage and mtROS production and/or oxidative stress. This can most likely be addressed by reminding the audience of the mitochondrial mutator mice, Trifunovic et al., 2005 PNAS, in which mice that incur high levels of mtDNA mutations are no different from WT mice in regards to protein oxidative damage and/or cell death.

8) In regards to Figure 1 and 2 (and supporting texts) the authors discuss the role that mtDNA has in immunity and NDD. In Figure 3, the authors introduce the concept that cancer cells transfer mtDNA from healthy cells to maintain survival of the cancer cells. I find this unclear, do the authors in the latter case mean that there is a transfer of whole mitochondria or simply just mtDNA. This must be clarified please.

9) In regards to cancer cells relying on the transfer of healthy mitochondria, why is there no signal from the healthy cell to prevent this transfer? How does the cancer cell overcome the healthy cell? Also in terms of a solid tumour, is it then only the outermost cancer cells that are capable of transferring healthy mitochondria?

I thank the authors in advance for addressing my concerns.   

Author Response

This review focuses on the emerging role of mitochondria in inflammation with connections to innate immunity and neurodegenerative disease and cancer. Thank you for the review, I think it is well organised and descriptive. Please see me few points for small areas of improvement.

R: We are grateful for the reviewer’s kind comments and valuable suggestions.

1) There are very few errors, but please have someone check for english grammar, this mainly pertains to singular/plural rules.

R: The reviewer is right. We have checked english grammar and we have corrected the numerous errors throughout the manuscript. The review has been also edited by a professional English service.

2) Line 97: Can the authors clarify what they mean by "...mitochondria are tolerated by immunity...". Do they mean that these organelles are not attacked by the immune system?

R: We agree with the reviewer, the sentence was not clear. According with other reviewer’s comments we re-organized the introduction and this sentence has been reformulated.

3) Lines 118-128: It is suggested that the authors strengthen their argument supporting the role of mtROS activating the inflammasome. Is this due to evidence that mitochondria are the main producers of ROS. In the same line, is it mtROS and/or mtDNA and/or pro-inflammatory cytokines that are activating the inflammasome. Is it the release of any of these mitochondrial components (or just one) that is activating the inflammasome.

R: We agree with the reviewer and we re-organized the introduction. Moreover, we better clarify the mechanisms that induce the NLRP3 inflammasome activation (lines 250-252, 269-273, 292-297).

4) Line 135-136 "...it is a great promise for treatment of different inflammatory diseases?" is not a question. R: The reviewer is right, likewise we decided to remove this sentence.

5) Regarding Figure 1, is viral RNA only targeting/influencing mitochondria (via MAVS) or this is a simplified figure? This should be clarified in the figure legend and or supported text. Also in regards to this figure, MAVS activation appears to be inducing the release of Nf-kB etc, is this release due to loss of membrane potential and the concomitant initiation of apoptosis?

R: We agree with the reviewer and we implemented Figure 1 adding more details on how virus or bacteria can infect mitochondria and alter innate immune response.

In a study by Koshiba T and colleagues it has been reported that cells defected in both mitofusins exhibited reduced mitochondrial membrane potential (MMP), impaired induction of interferons and proinflammatory cytokine, correlated with defective in MAVS-mediated antiviral signaling. The dissipation in MMP did not affect the activation of the transcription factor interferon regulatory factor 3 downstream of MAVS, which suggests that MMP and MAVS are coupled at the same stage in the RLR signaling pathway. However, if this dissipation of MMP can induce NF-kb release has not been directly demonstrated. A sentence has been added in the text to clarify the concept (lines 195-198).

6) Lines 226-228, the authors state that under "...resting conditions NLRP3 is in the cytosol but once activated it migrates to the mitochondria linked by MAMs...". Above, however, the authors write of the importance of the activation of the inflammasome due to mtROS, what then is "driving" the movement from the cytosol to the interaction with mitochondria, can it still be mtROS even if at rest/unstressed it is further away from the source of ROS? For me, this needs to be clarified.

R: We thank the reviewer for the comment. We have expanded the discussion of the mechanism that activate the NLRP3 inflammasome in the current version of the manuscript (lines 250-252, 269-273, 292-297).

7) Line 488, although there is no evidence suggesting that mtDNA interacts with histone proteins, mtDNA exists in nucleoids and interact with TFAM, polymerase gamma, twinkle, single stranded binding protein. This should be made more clear to the audience. In this line, it is recommended that the authors make more clear the evidence presented linking mtDNA damage and mtROS production and/or oxidative stress. This can most likely be addressed by reminding the audience of the mitochondrial mutator mice, Trifunovic et al., 2005 PNAS, in which mice that incur high levels of mtDNA mutations are no different from WT mice in regards to protein oxidative damage and/or cell death.

R: Thanks for the comment; we implemented the text with your suggested reference, discussing a bit more the role of mutation in mtDNA and ROS production (lines 581-588).

8) In regards to Figure 1 and 2 (and supporting texts) the authors discuss the role that mtDNA has in immunity and NDD. In Figure 3, the authors introduce the concept that cancer cells transfer mtDNA from healthy cells to maintain survival of the cancer cells. I find this unclear, do the authors in the latter case mean that there is a transfer of whole mitochondria or simply just mtDNA. This must be clarified please.

R: The legend of Figure 3 and the supporting text has been changed accordingly to the reviewer’s comment. Indeed, the sentence in the body of the review was misleading because Villanueva and collaborators reported that this transferring phenomenon was driven by the mitochondrion, while Tan and collaborators refer to mtDNA transfer in their research, most likely due to mitochondria transfer (never mentioning it though), nevertheless it has been clarified in the text by adding useful reference for a complete overview on what is known on this mechanism (lines 597-601).

9) In regards to cancer cells relying on the transfer of healthy mitochondria, why is there no signal from the healthy cell to prevent this transfer? How does the cancer cell overcome the healthy cell? Also in terms of a solid tumour, is it then only the outermost cancer cells that are capable of transferring healthy mitochondria?

R: The doubts raised by the reviewer are justifiable. Indeed, in the literature mining made for this review we were not able to find strong evidences on the cellular mechanism behind the mitochondria transfer from healthy to cancer cells, but rather more on a phenotypic rescue. It is still undefined how cancer cells can persuade surrounding cells to receive healthy mitochondria. Most of the in vitro and in vivo experiments were conducted on solid tumor cell lines; in many studies it has been noted that solid tumor cell lines were able to retrieve their respiratory capabilities taking mitochondria from MSCs and endothelial cells, possibly suggesting the participation of the circulating system in the process.

To date the knowledge on the topic refers to what has been stated in the review, we referred mostly to Berridge and collaborators review on the topic. Nevertheless, an explicative sentence has been added to the paragraph (lines 597-601).

We hope that the reviewer could appreciate the current version of the manuscript.

Reviewer 3 Report

Overall, this well-written review article captures the significance of mitochondria in the context of inflammation and diseases that results from the same. The authors have specifically focused on the role of mitochondrial dysfunction contributing to altered inflammatory response leading to neuro-degenerative diseases and cancer.

The authors provide a good overview of the current status of the field and present opportunities for future therapeutic strategies for reducing Reactive Oxygen Species. 

The figures illustrate the essence of the abstract adequately. 

Author Response

R: We thank the reviewer for his/her appreciation of the work

Reviewer 4 Report

The aim of the review submitted by Sonia Missiroli et al is to link mitochondria and inflammation. This review covers all the aspect of mitochondria in inflammation and its impact on diseases, such as neurodegenerative disease and cancer.

The review is divided in three parts:

  • Mitochondria: key players in innate immunity
  • Role of mitochondria and neuroinflammation in neurodegenerative diseases
  • Inflammation-related mitochondrial dysfunction in cancer: a negative loop.

The review is well written and the iconography is good and very supportive of the text.

Nevertheless, I have some suggestions that may clear the take home message.

In the Mitochondria: key players in innate immunity part, it would be better to subdivide into three paragraph with subheadings corresponding to the three part of the figure 1. For instance, 1- viral infection, 2- Inflammasome and 3- bacteria infection in that order. Moreover, the part on bacteria infection should be balanced in term of size with the two other parts. Finally, a small paragraph on MAMs may be added

The part on the therapeutic aspect should be separated from the cancer paragraph with a related figure. May be figure 3 could be cut in half.

Author Response

The aim of the review submitted by Sonia Missiroli et al is to link mitochondria and inflammation. This review covers all the aspect of mitochondria in inflammation and its impact on diseases, such as neurodegenerative disease and cancer.

The review is divided in three parts:

  • Mitochondria: key players in innate immunity
  • Role of mitochondria and neuroinflammation in neurodegenerative diseases
  • Inflammation-related mitochondrial dysfunction in cancer: a negative loop.

The review is well written and the iconography is good and very supportive of the text.

Nevertheless, I have some suggestions that may clear the take home message.

In the Mitochondria: key players in innate immunity part, it would be better to subdivide into three paragraph with subheadings corresponding to the three part of the figure 1. For instance, 1- viral infection, 2- Inflammasome and 3- bacteria infection in that order. Moreover, the part on bacteria infection should be balanced in term of size with the two other parts. Finally, a small paragraph on MAMs may be added

R: We thank the reviewer for the constructive comments and valuable suggestions that strengthen our manuscript. As suggested we have re-organized the part “mitochondria: key players in innate immunity” and we subdivided into three paragraphs (lines 174, 241 and 307) The part on bacterial infection has been improved both in the text and in the figure (lines 307-345). As requested, a small paragraph on MAMs has been added (lines 260-267)

The part on the therapeutic aspect should be separated from the cancer paragraph with a related figure. May be figure 3 could be cut in half.

R: We appreciated the reviewer’s suggestion, but since we discuss only two inflammatory diseases we prefer to maintain the therapeutic aspect within the paragraph for each pathology (lines 471-494 and lines 657-681), in order to better emphasize the current therapeutic directly connected to the disorder. We hope that the reviewer could appreciate the current version of the manuscript.